# Cognitive Performance Following Ingestion of Glucose–Fructose Sweeteners That Impart Different Postprandial Glycaemic Responses: A Randomised Control Trial

**DOI:** 10.3390/nu11112647

**Published:** 2019-11-04

**Authors:** Celeste Keesing, Brianna Mills, Charlene Rapsey, Jillian Haszard, Bernard Venn

**Affiliations:** 1Department of Human Nutrition, University of Otago, PO Box 56, Dunedin 9054, New Zealand; celestekeesing@gmail.com (C.K.); briannamills@hotmail.co.nz (B.M.); jill.haszard@otago.ac.nz (J.H.); 2Department of Psychological Medicine, Otago Medical School, University of Otago, PO Box 56, Dunedin 9054, New Zealand; charlene.rapsey@otago.ac.nz

**Keywords:** glucose, fructose, sucrose, isomaltulose, glycaemia, insulinaemia, cognition

## Abstract

We aimed to investigate the isolated effect of glycaemia on cognitive test performance by using beverages sweetened with two different glucose–fructose disaccharides, sucrose and isomaltulose. In a randomised crossover design, 70 healthy adults received a low-glycaemic-index (GI) isomaltulose and sucralose beverage (GI 32) and a high-GI sucrose beverage (GI 65) on two occasions that were separated by two weeks. Following beverage ingestion, declarative memory and immediate word recall were examined at 30, 80 and 130 min. At 140 min, executive function was tested. To confirm that the glycaemic response of the test beverages matched published GI estimates, a subsample (*n* = 12) of the cognitive testing population (*n* = 70) underwent glycaemic response testing on different test days. A significantly lower value of mean (95% CI) blood glucose concentration incremental area under the curve (iAUC) was found for isomaltulose, in comparison to the blood glucose concentration iAUC value for sucrose, the difference corresponding to −44 mmol/L∙min (−70, −18), *p* = 0.003. The mean (95% CI) difference in numbers of correct answers or words recalled between beverages at 30, 80 and 130 min were 0.1 (−0.2, 0.5), −0.3 (−0.8, 0.2) and 0.0 (−0.5, 0.5) for declarative memory, and −0.5 (−1.4, 0.3), 0.4 (−0.4, 1.3) and −0.4 (−1.1, 0.4) for immediate free word recall. At 140 min, the mean difference in the trail-making test between beverages was −0.3 sec (−6.9, 6.3). None of these differences were statistically or clinically significant. In summary, cognitive performance was unaffected by different glycaemic responses to beverages during the postprandial period of 140 min.

## 1. Introduction

Cognitive performance is a key factor in learning, and glucose is the main energy source of the brain [1]. This relationship between cognition and fuel source has resulted in trials comparing the effects of carbohydrates of differing predicted glycaemic responses on cognitive performance [2,3]. The findings of such trials have been inconsistent, both internally—in the same study, different associations across a battery of tests or differences for the same test that was carried out at different times—and externally, between studies. For example, some aspects of cognitive tests were found to be better with low-, compared to high-, glycaemic-index (GI) or glycaemic-load (GL) foods [3,4,5,6,7]. Other aspects were found to be better with high-, compared to low-, GI foods [3,8,9]. However, some studies found no association between the GI or GL of a test food and cognitive performance [2,7,9].

Inconsistencies in these findings may be due to differences in study design and methods to control for confounding factors and bias. In many of these studies, confounding factors, such as baseline cognitive performance [10], macro- and micronutrient content [2], energy [11], or palatability of the test foods [12], were not controlled. The use of whole foods may mean that a study lacks blinding, which may create a problem since cognitive tests are subjective and possess the potential to be influenced by investigators’ or participants’ opinions about one test food or beverage [13]. Differences in cognitive function have been found in many unblinded studies [2,3,4,5,6,8,11,12,13]. Variation in nutrients and energy between test foods may also affect cognitive performance [14,15,16]. A higher protein intake was found to be associated with better short-term memory [14], and greater accuracy was associated with a higher dietary fibre intake in comparison to control test foods [16]. In some studies, differences in cognition have been attributed to assumed (untested) differences in glycaemic response or to glycaemic responses that were not significantly different at cognitive test times [7,10,13]. All of these study design factors could influence the cognitive test outcomes independently of any differences in glycaemic response. Therefore, it is important to control for energy and macro- and micronutrient content and to ensure blinding and testing of the glycaemic response. The time of day of the cognitive testing is also relevant. Testing has been mainly undertaken after a morning fast However, students also have afternoon classes that require good cognitive functioning. Indeed, it has been found that university students perform better cognitively with afternoon, rather than morning, start times [17], but little research has been conducted on the effects of different glycaemic levels in the afternoon.

The aim of this study was therefore to isolate the effect of the glycaemic response on cognitive performance from potential sources of confounding and bias in a simulation of an afternoon lecture by providing sweetened beverages controlling for taste, appearance, volume, energy, and micro and macronutrient content. Our hypothesis was that cognitive test scores would be better when blood glucose concentrations were high than when they were low. Confirmation of differences in the glycaemic response to two different sweeteners was tested in a subset of participants. The primary outcome was the results of cognitive testing.

## 2. Materials and Methods

The University of Otago Human Ethics Committee (health) granted ethical approval for this study in October 2017 (Ethics committee reference number 17/011). The trial was registered at Australian and New Zealand Clinical Trials (ACTRN12618000901202).

### 2.1. Study Design

The study had a randomised double-blind (RCT) crossover design. Seventy-seven Otago University students aged 18–60 years were recruited. The order in which participants received the drinks was randomised and stratified by visit using random length blocks in Stata 15.1 (StataCorp, College Station, TX, USA). Allocation and exclusion information is shown in Figure 2. Exclusion criteria were sensitivities to artificial sweeteners or unwillingness to eat sushi. Prior to the main trial, the beverages underwent sensory testing by six people uninvolved in the main trial in order to match sweetness and taste using a blind randomised triangle sensory test [18]. Glycaemic responses to the two test beverages were measured in a subset of 12 of the 77 participants on separate test days.

### 2.2. Sweetened Beverages

The beverages (500 mL) were made in-house by adding 50.0 ± 0.01 g of sucrose (Caster Sugar 172323, Smart Choice, New Zealand) or isomaltulose (Palatinose^®^, Myprotein, Cheschire, United Kingdom) to carbonated water (Pure NZ sparkling water, NZ drinks Ltd., Pokeno, New Zealand). In order to adjust for sweetness, the isomaltulose beverage contained 0.035 g of sucralose (98% sucralose powder, J66736, lot: T21D050 Alfa Aesar; Fengxian, Shanghai, China). Both test beverages contained 50 µL of lemon flavour (Lemon 59223, lot:1002802470, Invita NZ Ltd.. Auckland, New Zealand). Sugars and sucralose were weighed using calibrated electronic scales (Sartorius 0.01/0.1 mg, Göttingen, Germany). The unmarked beverages were placed on separate trays, and a university staff member, who was otherwise uninvolved in the study, coded the beverages.

### 2.3. Test Day Procedures

The same procedures were used for each glycaemic test day (*n* = 12) and for each cognitive test day (*n* = 77). Test days, each separated by one week, involved participants fasting from 10:00 to 12:00. At 12:00, a lunch comprising water and eight pieces of commercially purchased maki sushi was provided to the participants in order to standardize the pre-test conditions. After eating, the participants could leave the facility with instructions not to eat anything else or undertake vigorous physical activity until their return at 13:45. At 14:00, the participants consumed the coded test beverage within 10 min. Participants and investigators were blinded to the type of sweetened beverage that was allocated. Following ingestion of the test beverage, cognitive testing took place over three hours. 

### 2.4. Glycaemic and Insulinaemic Response Testing

The subset of 12 participants had 500 µL capillary blood samples taken using a contact-activated 1.5 mm × 2.0 mm disposable lancet (Becton Dickinson Microtainer^®^, Franklin Lakes, NJ, USA). Samples were collected at 13:50, prior to consumption of the sweetened beverages, and following ingestion at 30, 60, 90, 120, 150 and 180 min. Glucose was analysed on a Roche/Hitachi Cobas c311 (Roche, Indianapolis, IN, USA) using an enzymatic colorimetric method. Insulin was analysed using an electrochemiluminescence immunoassay on a Roche/Hitachi Cobas e411 (Roche, Indianapolis, IN, USA). 

### 2.5. Cognitive Testing

It has been suggested that the use of film watching to assess cognition is an underutilized area of research [19]. Following the ingestion of the test beverage, cognitive testing was therefore undertaken by displaying a film in 30-min segments. At 30, 80 and 130 min, a set of film questions and a word recall were administered. At 140 min, Reitan’s Trail Making Part B was undertaken.

Film recall questions: The BBC Horizon documentary “Sugar vs. Fat” was screened on the first cognitive test day, while “That Sugar Film” (Madman Entertainment, Melbourne, Australia) was screened on the second day. In order to test declarative memory, the participants filled in a 10-item questionnaire relating to the 30-min segment that had been most recently viewed. The questionnaires given at 30, 80 and 130 min consisted of three multi-choice questions, four auditory written answer questions, and three visual written answer questions. 

Immediate word recall: These tests were carried out after the film recall. A 25-word wordlist was read out at one word every two seconds. On each cognitive test day, three wordlists were used (six wordlists in total over two test days). The wordlists included standardised and validated word categories [20,21]. 

Reitan’s Trail Making: This is a widely used test [22], requiring participants to join alternating numbers and letters in numerically ascending and alphabetical order (e.g., 1—A to 2—B, up to the letter L) without lifting the pen off the paper. The investigators timed individuals, with the outcome measure being time in seconds to completion. The mirror image of Reitan’s Trail Making Part B was used on the second test day.

A schematic of the blood sampling and cognitive test scheduling is given in Figure 1.

### 2.6. Statistical Analysis

A sample size of 60 would be adequate to detect a difference of 0.5 SD for all cognitive tests in standardised outcomes (α = 0.01 level with 90% power). Using in-house glycaemic data, a sample size of 12 would be sufficient to detect a 33% change in glycaemic incremental area under the curve (iAUC) using the 5% level of significance with 80% power. Differences in cognitive test scores between test beverages were calculated at 30, 80 and 130 min using mixed-effects regression with the participant as a random effect. Participants with incomplete data or who did not consume the beverage on a test day were excluded from the analysis.

## 3. Results

Recruitment occurred in February 2018, and testing commenced during March 2018. The flow of participants is given in Figure 2.

### 3.1. Sensory Testing

The sensory testing (*n* = 6) indicated that the test beverages were indistinguishable in sweetness and taste.

### 3.2. Glycaemic and Insulinaemic Testing

The 12 participants who participated in the glycaemic testing of the sugary beverages had a mean (SD) age of 21.2 (1.4) years and a body mass index of 21.9 (3.7) kg/m^2^. The mean (SD) iAUC was 61 (52) mmol/L∙min for the sucrose beverage and 17 (52) mmol/L∙min for the isomaltulose beverage, with a statistically significant mean (95% CI) difference of 44 mmol/L∙min (18, 70). The corresponding insulin incremental areas under the curve were 2492 (1540) and 609 (1115) μIU/L∙min for the sucrose and isomaltulose beverages, respectively, with a mean difference of 1883 μIU/L∙min (921, 2845). The mean difference in blood glucose concentration between the test drinks at 30 min was 1.6 (95% CI: 1.2, 2.1) mmol/L, and that for insulin was 47 (95% CI: 23, 71) μIU/L. The concentrations of these factors were not statistically significantly different at the 90- and 120-min timepoints. Plots of the mean glycaemic and insulinaemic responses are shown in Figure 3.

### 3.3. Cognitive Performance Testing

The 70 participants who completed the main trial were predominantly female (*n* = 57 out of 70) with a mean (SD) age of 21.9 (0.64) years and a body mass index of 23.3 (2.7) kg/m^2^. Ethnically, 62% were of European descent, 4% Maori, 12% Chinese and 11% other. The adjusted effects of isomaltulose and sucrose on cognitive tests are presented in Table 1.

There were no significant differences in treatment effect between the sugars for declarative memory (film questions), immediate free recall (word recall), or executive function (Reitan’s Trail Making Part B) performance. No participants were able to recall all 25 words in the word recall tests.

## 4. Discussion

There were confirmed differences in the metabolic responses following ingestion of the test beverages. Unexpectedly, the glycaemic and insulinaemic responses to the isomaltulose beverage declined relative to baseline. We are unaware that this has been observed previously, although it is known that glycaemic responses are dampened during the day compared to following an overnight fast [23]. Despite the differences in glycaemic and insulinaemic responses, there were no statistically significant differences in cognitive test scores following the ingestion of glucose–fructose sugar-sweetened beverages that induced different postprandial glycaemic and insulinaemic responses. The finding of no difference in cognitive test scores during an afternoon postprandial period are novel. In previous work, memory was better in the afternoon following a glucose, compared to an aspartame-sweetened, drink [24]. However, differences in study design could have accounted for the different results, as the drinks that were used by Sünram-Lea et al. were not isocaloric, and testing was done starting two hours postprandially [24]. By contrast, our drinks were isocaloric, and testing was started 30 min postprandially and at a time when glycaemic differences between drinks were maximal. Our results are consistent with some studies on adult populations in which no differences in tests of free recall, declarative memory or executive function were found in the morning. For example, in a crossover study in which breakfasts having considerably different GI (64.0 and 29.4) were provided to young adults, no between-treatment difference in verbal memory, phonological fluency or attention and executive functions were found over the postprandial period [2]. In contrast, some differences in cognitive test scores in the morning have been found following manipulation of the glycaemic response. Using a parallel study design, 36 female university students who ate a lower GI meal (GI 42.3) had better recollection of words at 150 and 210 min than 35 students who ate a higher GI meal (GI 65.9), with no between-treatment difference at 30 or 90 min [11]. Although insulin concentration was not measured in that study, the authors speculated that a lower GI generates a smoother insulin profile than a higher GI, resulting in less glucose uptake by insulin-sensitive tissues leaving more for brain use in the later postprandial phase [11]. Our data support a GI-dependent difference in insulin concentration profiles over the postprandial period, even though in our study, the between-treatment insulin concentrations were not different at the 150- and 180-min timepoints. Inconsistency in findings among studies of GI and cognitive functioning have been reviewed, with the authors recommending that methodologic problems, such as the use of standardized tests, matching treatments on energy and macronutrient composition, standardizing pre-test conditions, recruiting a homogenous sample, and limiting the number of domains, should be addressed [25]. Many domains have been tested and found to be affected by very low blood glucose concentration (hypoglycaemia ≤ 47 mg/dL or 2.6 mmol/L) [26]. However, hypoglycaemia was not present in the postprandial period of our healthy participants, and from the review carried out by Phillipou et al. there is no consistency in the association between postprandial glycaemia and any particular cognitive domain [25]. Indeed, inconsistent results among studies may be due to heterogeneity in design, which also impairs direct comparisons. Variations in study design include type of cognitive tests, timing of assessment [3,5,13], blinding protocols [3,5,6,8,27], methods of measuring or inferring glycaemic response [3,10,12] and type of test foods [2,3,5,6,11,27,28]. If the hypothesis being tested is that cognition is related to glycaemia, then there is an argument that the test foods should have the same macronutrient and fibre content. This is because cognitive performance has been found to be influenced by protein and fibre [14,16].

Our hypothesis that cognitive test scores would be better when blood glucose concentrations were higher, rather than lower, was not upheld. An explanation may be the neuroendocrine control of glucose allocation through which the brain is the first glucose receiver ahead of all other glucose-requiring organs, with glucose supply to the brain being insulin-independent [29]. Glucose supply to the brain is a continuous physiological priority [30], suggesting that brain function is independent of postprandial fluctuations in blood glucose concentration.

A morning study with comparable study design elements to this study included a water control and matching isomaltulose- and sucrose-sweetened milky beverages matched on volume, macronutrient content, and taste. These beverages were given to participants in randomized order [9]. The results of visual verbal learning tests (immediate recall, delayed recall, recognition of correct responses and response time) and psychomotor tests (reaction time and correct responses) were not different among beverages. The only difference in cognitive performance was a better outcome for Serial Sevens after ingestion of the sucrose-sweetened drink [9]. As milk protein is insulinotropic [31], the authors commented that this may have influenced the glycaemic response, suggesting that the sweeteners should be used in a neutral vehicle. Hence, our data are consistent with those of Dye et al., who interpreted their data to refute the notion that postprandial glycaemia was associated with cognitive performance [9].

A potential limitation of our study design is the use of sucralose added to the isomaltulose beverage in order to match the sweetness of the sucrose beverage. Sucralose has been found to activate taste pathways in the brain differently from sucrose [32]. Whether this differential in taste pathways could affect memory is unclear, although when a sucralose or water protocol was administered to mice for 32 days, there was no difference in memory retention between treatments [33]. In addition, the amount of sucralose used in our study was small (approximately 0.5 mg/kg body weight) in comparison to the amount used for the mice, which were given the maximum allowable dose of 16,000 mg/kg body weight [33]. Thus, it seems unlikely that sucralose would have had an effect on cognition in our participants.

A strength of our research is the use of film watching. This approach has been used previously in studies of cognition [19,34,35] but not in combination with glycaemic studies. This aspect adds novelty to our findings. We used film watching to simulate a lecture setting for the students that also involved visual and auditory information processing [36,37]. A possible limitation of the method is that the degree of interest may have differed between films shown on different days. However, so as to minimise differences, the order in which the students received the beverages was randomised. Other strengths of our research include matching the sweetness of the test beverages to ensure true blinding of the participants, randomisation of treatment order and blinding of the investigators, who were became aware of the treatment code only after data analysis. Our findings indicate that cognitive performance following the ingestion of beverages containing sugars differing in GI is independent of the postprandial glycaemic response. 

## 5. Conclusions

Our data do not support a relationship between postprandial differences in glycaemia and cognitive performance.

## Figures and Tables

**Figure 1 nutrients-11-02647-f001:**
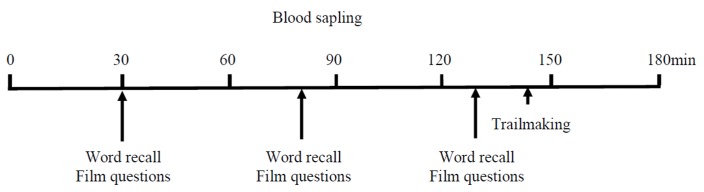
Timing of the blood sampling (0–180 min) and cognitive test times relative to baseline (0 min).

**Figure 2 nutrients-11-02647-f002:**
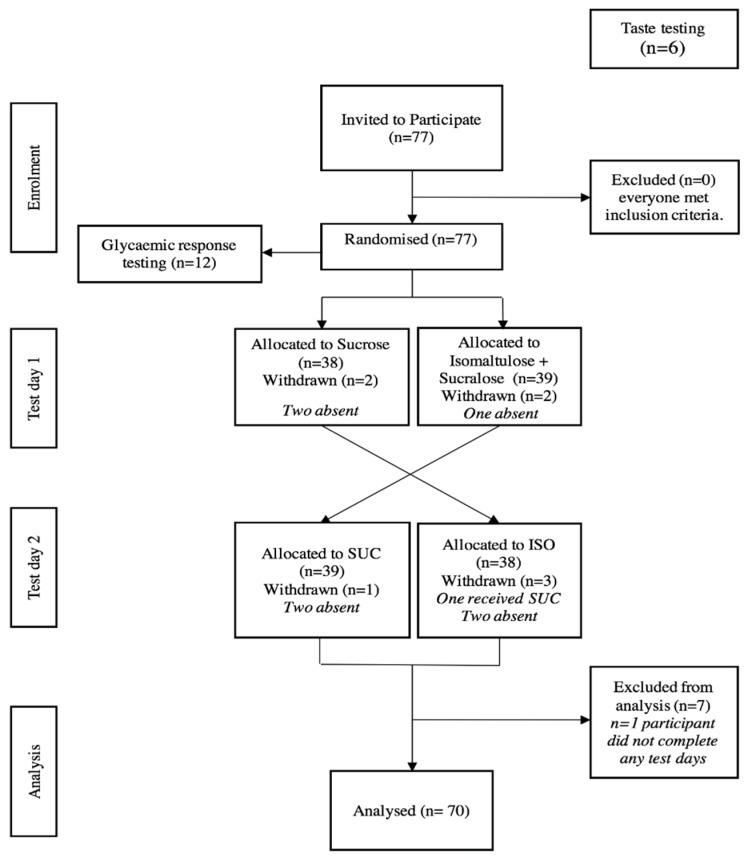
Study design and participant flow. SUC = sucrose; ISO = isomaltulose and sucralose.

**Figure 3 nutrients-11-02647-f003:**
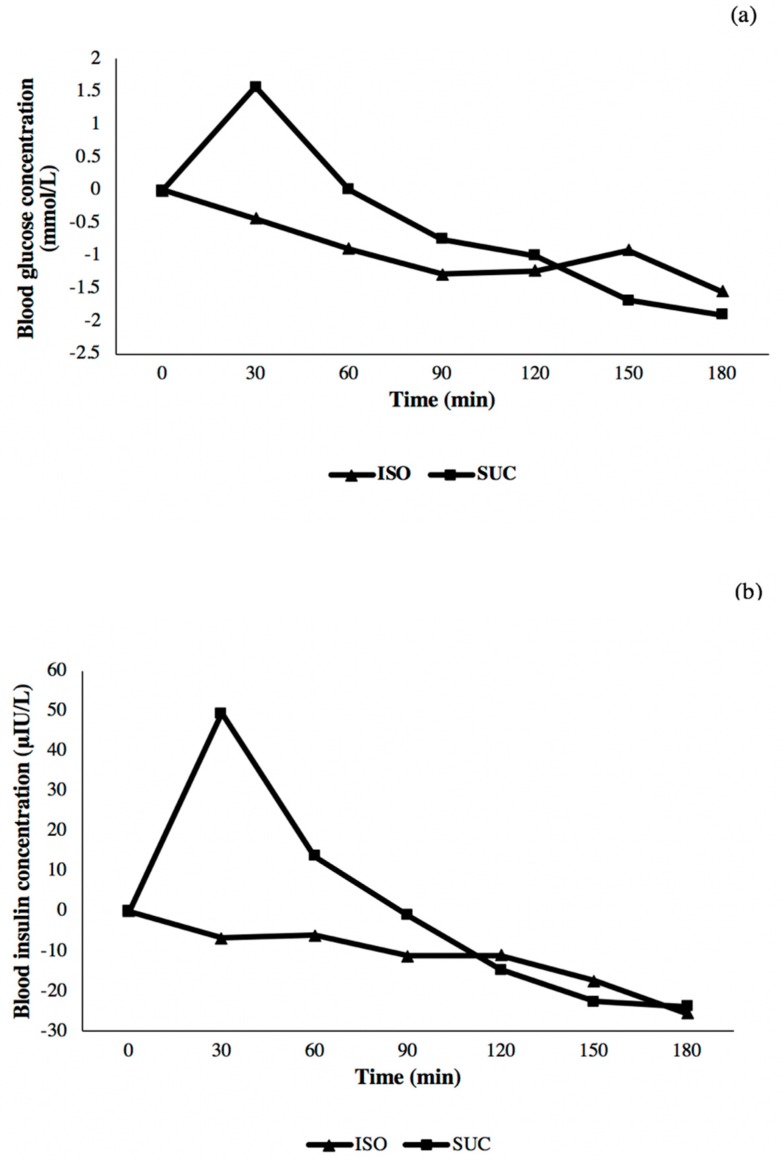
Incremental mean blood glucose (**a**) and insulin responses (**b**) in 12 participants to ISO and SUC beverages.

**Table 1 nutrients-11-02647-t001:** Adjusted mean (SD) difference of cognitive test results between sugars over three hours.

Cognitive Test	Time (min)	Sucrose (*n* = 70)	Isomaltulose + Sucralose (*n* = 70)	Mean Difference (95% CI)	*p*
Film Question (*n*)	30	5.6 (1.5)	5.6 (1.4)	0.1 (−0.2, 0.5)	0.463
Film Question (*n*)	80	5.5 (1.8)	5.2 (1.8)	−0.3 (−0.8, 0.2)	0.252
Film Question (*n*)	130	5.0 (1.5)	5.0 (1.7)	0.0 (−0.5, 0.5)	0.927
Word Recall (*n*)	30	11.4 (3.4)	10.7 (3.0)	−0.5 (−1.4, 0.3)	0.198
Word Recall (*n*)	80	10.7 (3.2)	11.1 (3.5)	0.4 (−0.4, 1.3)	0.301
Word Recall (*n*)	130	11.0 (3.5)	10.5 (3.3)	−0.4 (−1.1, 0.4)	0.357
Trail Making Part B (sec)	140	52.3 (26.1)	53.0 (23.9)	−0.3 (−6.9, 6.3)	0.928

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
