# Peer review of "Cognitive Performance Following Ingestion of Glucose–Fructose Sweeteners That Impart Different Postprandial Glycaemic Responses: A Randomised Control Trial"

_nutrients, 2019, doi:10.3390/nu11112647_

Round 1

Reviewer 1 Report

The aim of the present study was to investigate the isolated effect of glycaemia on cognitive test performance by using beverages sweetened with sucrose or isomaltulose. in general, the study was carefully conducted and the manuscript was well prepared. There are several major concerns:

The rationale of the present study is weak. Why choosing afternoon to test the glycaemic responses and cognitive function? What is the hypothesis? Any evidence to support the hypothesis?  In discussion section, what is the new finding of the present study? What may be the possible mechanisms behind the results? Further discussion is definitely needed.  It seems that the present study was similar to one previous study (ref 9). What are the similarities and differences between these two studies? Further discussion is needed. What is the conclusion?

Some minor concerns:

Page 1, line 42, Ref 13? Format is different with other references. Please check. Page 2, line 69, students aged between 18-60 yrs?  Page 2, line 90, all participants consumed the same lunch without considering age, gender, body weight, etc.? Page 3, line 108, ref to support the film test? Page 3, line 118, ref to support the trail making test? Page 5, figure 2, glucose use incremental glucose concentrations while insulin use insulin concentrations?

Author Response

We would like to thank the reviewer for helpful comments that we hope are now addressed.

Why choosing afternoon to test the glycaemic responses and cognitive function? What is the hypothesis? Any evidence to support the hypothesis? 

We have added the following text into the Introduction. It has been found that university students perform better cognitively following afternoon compared with morning start times {Evans, 2017 #639}, but there has been little research conducted comparing the effects of manipulating glycaemia in the afternoon. Our hypothesis is that cognitive test scores will be better when blood glucose concentrations are higher compared with lower.

In discussion section, what is the new finding of the present study?

We have added the following into the discussion. The findings of no difference in cognitive test scores during an afternoon postprandial period are novel. In previous work, memory was better in the afternoon following a glucose compared with an aspartame-sweetened drink {Sünram-Lea, 2001 #312}. However, differences in study design could have accounted for the different results as the drinks that were used by Sunram-Lea et al were not isocaloric and testing was done starting two hours postprandially {Sünram-Lea, 2001 #312}, whereas our drinks were isocaloric and testing was started 30 min postprandially at a time when glycaemic differences between drinks were maximal. Our results are consistent with some studies on adult populations in which no differences in tests of free recall, declarative memory or executive function were found in the morning {Dye, 2010 #121;Sanchez-Aguadero, 2018 #309;Young, 2014 #179;Nilsson, 2009 #123}.

What may be the possible mechanisms behind the results?

Thank you for this comment, we have added the following to the discussion. Our hypothesis that cognitive test scores would be better when blood glucose concentrations were higher compared with lower was not upheld. An explanation may be the neuroendocrine control of glucose allocation, with glucose supply to the brain being insulin-independent facilitating the brain as first receiver ahead of all other glucose-requiring organs {Peters, 2002 #640}. Glucose supply to the brain is a continuous physiological priority {Ritter,  #641}, suggestive that brain function is independent of postprandial fluctuations in blood glucose concentration.

It seems that the present study was similar to one previous study (ref 9). What are the similarities and differences between these two studies? Further discussion is needed. What is the conclusion?

In the study by Dye et al, sweetened milk drinks were used. The protein in milk is insulinotropic whereas we wanted to isolate the glycaemic effects of the sweeteners to a difference in the rate of carbohydrate digestion alone. We have added the following into the discussion: “As milk protein is insulinotropic {Nilsson, 2004 #642}, the authors commented that this may have influenced the glycaemic response, suggesting that the sweeteners be used in a neutral vehicle. Hence, our data are consistent with those of Dye et al who interpreted their data to refute the notion that postprandial glycaemia was associated with cognitive performance {Dye, 2010 #121}”. This is consistent with our conclusion.

Some minor concerns:

Page 1, line 42, Ref 13? Format is different with other references. Please check.

Corrected.

Page 2, line 69, students aged between 18-60 yrs? 

Yes, we do have mature students in our classes.

Page 2, line 90, all participants consumed the same lunch without considering age, gender, body weight, etc.?

Yes, this was a crossover trial so there was within-person control.

Page 3, line 108, ref to support the film test?

We have added a reference and the following text: It is suggested that the use of film to assess cognition is an underutilized area of research [19]. Cognitive testing was therefore undertaken following ingestion of the test beverage after which a film was displayed in 30-minute segments.

Page 3, line 118, ref to support the trail making test?

We have added a reference (now line 135)

Page 5, figure 2, glucose use incremental glucose concentrations while insulin use insulin concentrations?

Thank you, we have made this consistent by removing ‘incremental’ from the glucose axis and adding ‘incremental’ into the figure legend.

Reviewer 2 Report

In this study, authors examined the impact of postprandial glycaemia on cognitive performance through a randomized control trial with intake of sucrose or isomaltulose.

Although the major conclusion of this study has been demonstrated in several study, the findings in this study remains to provide some information to readers. However, there are some points needed to be concerned.   

A more detailed description to explain study design is suggested, such as the time points for blood samplings and the difference between the test of day 1 and day 2. It seems that there is no difference in glycaemia and insulin concentration between two groups at the time point of word recall test and Trail making. A description or table for time correlation between glycaemia and each test should be added in the manuscript.

Author Response

We would like to thank the reviewer for helpful comments that we hope are now addressed.

A more detailed description to explain study design is suggested, such as the time points for blood samplings and the difference between the test of day 1 and day 2.

In response to Reviewer 1 we have added text to support the rationale for the design into the Introduction (lines 57 to 60 and 64 to 65). We collected blood at regular 30 min intervals. The first 30 min sample was critical as from previous work we had expected this timepoint to coincide with the peak response. The following timepoints were used to plot the glycaemic and insulinaemic response over time. Again, from previous work we have found that the change in the blood concentration of these factors is predictable and that 30 min intervals is sufficient to characterise differences in trajectory between the treatments. The cognitive test days 1 and 2 followed an identical protocol.

It seems that there is no difference in glycaemia and insulin concentration between two groups at the time point of word recall test and Trail making.

Word recall was carried out at the 30, 80 and 130 min timepoints. There was a between-treatment difference in glucose and insulin concentration at the 30 min timepoint. We have added this into the Results section (lines 173-6). The time courses for change in glucose and insulin concentration; and the times for administering the cognitive tests; are typical of this literature. Although differences in blood concentration may be negligible at later timepoints, people have still tested at these times on the hypothesis that a slower decline in concentration might be of cognitive benefit by providing ‘longer lasting energy’.

A description or table for time correlation between glycaemia and each test should be added in the manuscript.

We have added a schematic to the Methods

Round 2

Reviewer 1 Report

The authors have addressed most of my concerns, and the quality of paper has improved. I still have one concern about the discussion part. The purpose of the present study was to investigate the effect of glycaemia on cognitive test performance. Therefore, the discussion should focus more on the effect of glycaemia (or glycemic index?). In the current version, the author added one reference (ref 24) to support the discussion. However, this study compared the glucose drink and aspartame-sweetened drink. it is more like the comparison of glucose vs. non-CHO. Also, the authors may try to discuss the effect of glycaemia on different domains of cognitive function, rather than simply attribute the inconsistent results in previous studies to heterogeneity in study design. 

Author Response

The purpose of the present study was to investigate the effect of glycaemia on cognitive test performance. Therefore, the discussion should focus more on the effect of glycaemia (or glycemic index?). In the current version, the author added one reference (ref 24) to support the discussion. However, this study compared the glucose drink and aspartame-sweetened drink. it is more like the comparison of glucose vs. non-CHO. Also, the authors may try to discuss the effect of glycaemia on different domains of cognitive function, rather than simply attribute the inconsistent results in previous studies to heterogeneity in study design. 

Thank you for the comment. We have added in the following text into the discussion that we hope addresses the comments. “In a crossover study in which breakfasts having considerably different GI (64.0 and 29.4) were provided to young adults, no between-treatment difference in verbal memory, phonological fluency or attention and executive functions were found over the postprandial period [2]. In contrast, some differences in cognitive test scores in the morning have been found following manipulation of the glycaemic response. Using a parallel study design, 36 female university students who ate a lower GI meal (GI 42.3) had better recollection of words at 150 and 210 min than 35 students who ate a higher GI meal (GI 65.9), with no between-treatment difference at 30 or 90 min [11]. Although insulin concentration was not measured in that study, the authors speculated that lower GI generates a smoother insulin profile than high GI resulting in less glucose uptake by insulin-sensitive tissue leaving more for brain use in the later postprandial phase [11]. Our data support a GI-dependent difference in insulin concentration profiles over the postprandial period although in our study, the between-treatment insulin concentrations were not different at the 150 and 180 min timepoints. Inconsistency in findings among studies of GI and cognitive functioning have been reviewed, with the authors recommending that methodologic problems are addressed including the use of standardized tests, matching treatments on energy and macronutrient composition, standardizing pre-test conditions, recruiting a homogenous sample, and limiting the number of domains [25]. Many domains have been tested and found to be affected by very low blood glucose concentration (hypoglycaemia ≤ 47mg/dL or 2.6mmol/L) [26]. However, hypoglycaemia was not present in the postprandial period of our healthy participants and from the review carried out by Phillipou et al there is no consistency in association between postprandial glycaemia and any particular cognitive domain [25].”

Reviewer 2 Report

I have no further comments.

Author Response

Thank you for the time taken to review our manuscript